# Targets in the Tumour Matrisome to Promote Cancer Therapy Response

**DOI:** 10.3390/cancers16101847

**Published:** 2024-05-11

**Authors:** Siti Munira Abd Jalil, Jack C. Henry, Angus J. M. Cameron

**Affiliations:** Barts Cancer Institute, Queen Mary University of London, John Vane Science Centre, Charterhouse Square, London EC1M 6BQ, UK; s.abdjalil@qmul.ac.uk (S.M.A.J.); jack.henry@qmul.ac.uk (J.C.H.)

**Keywords:** extracellular matrix, tumour matrisome, tumour microenvironment, anti-tumour immunity, targeted therapy, immunotherapy, stroma, cancer-associated fibroblasts

## Abstract

**Simple Summary:**

The extracellular matrix acts as scaffolding to support the structure and function of the cells in our body. In cancer, this matrix is altered in a way that helps cancer cells to grow, spread and avoid the immune system. The altered matrix can also prevent cancer treatments like chemotherapy and immunotherapy from working. Targeting the matrix with drugs is emerging as an exciting way to modify the scaffold in a precise way to improve the effectiveness of cancer treatments in patients. In this review, we will examine strategies to target the matrix with drugs, which can help the immune system fight cancer and improve the response to existing cancer therapies.

**Abstract:**

The extracellular matrix (ECM) is composed of complex fibrillar proteins, proteoglycans, and macromolecules, generated by stromal, immune, and cancer cells. The components and organisation of the matrix evolves as tumours progress to invasive disease and metastasis. In many solid tumours, dense fibrotic ECM has been hypothesised to impede therapy response by limiting drug and immune cell access. Interventions to target individual components of the ECM, collectively termed the matrisome, have, however, revealed complex tumour-suppressor, tumour-promoter, and immune-modulatory functions, which have complicated clinical translation. The degree to which distinct components of the matrisome can dictate tumour phenotypes and response to therapy is the subject of intense study. A primary aim is to identify therapeutic opportunities within the matrisome, which might support a better response to existing therapies. Many matrix signatures have been developed which can predict prognosis, immune cell content, and immunotherapy responses. In this review, we will examine key components of the matrisome which have been associated with advanced tumours and therapy resistance. We have primarily focussed here on targeting matrisome components, rather than specific cell types, although several examples are described where cells of origin can dramatically affect tumour roles for matrix components. As we unravel the complex biochemical, biophysical, and intracellular transduction mechanisms associated with the ECM, numerous therapeutic opportunities will be identified to modify tumour progression and therapy response.

## 1. Matrix Signatures Are Prognostic and Predict Immune Content

Deregulation of the tumour matrisome is a characteristic of all solid tumours, where an aberrant matrix can dictate the hallmarks of cancer [1]. The matrix is composed of a complex network or fibrillar proteins, proteoglycans, and other macromolecules, which are intricately crosslinked and subjected to post-translational modification. The term matrisome is used to describe an inventory of genes or proteins, which form the matrix or regulate matrix composition, structure, or function. In mammals, there are approximately ~300 core matrisome components (e.g., NABA_CORE_MATRISOME gene set [2]) and a large number of matrix-associated proteins and enzymatic regulators, which are variously included within the broad definition of the matrisome [3]. Approximately 1000 genes encoding matrix or matrix-associated proteins are commonly included in matrisome gene ontology lists [4]. The matrix provides the scaffold in which tissues are embedded and also directs cellular behaviour through direct interaction with numerous cellular receptors, tissue biomechanics, the regulation of growth factor availability, and physical restriction [5]. Tissue organisation governed by the matrix also critically regulates how and when distinct cell types interact with each other. In cancer, this is particularly pertinent to stromal interactions, tumour vascularisation, and immune landscape [6].

Matrix deregulation in solid cancers is perhaps most apparent as tumour desmoplasia, where excessive fibrotic deposition of the ECM leads to tissue stiffening. In many aggressive tumours, including PDAC, breast cancers, and ovarian cancer, desmoplasia is a defining feature, which often correlates with poor outcome and treatment failure [7,8,9,10,11,12]. However, unravelling the correlation of desmoplasia with causative roles in aggressive disease has proven more difficult. Supporting a causative role, tissue stiffness and fibrosis are well-known risk factors for several cancers. Mammographic density identified through population screening is among the highest risk indicators for developing breast cancer [13,14]. Furthermore, fibrotic conditions, such as liver cirrhosis and pancreatitis, are associated with significant elevated cancer risk [15,16,17]. Nonetheless, interventions to modify tumour fibrosis have had mixed outcomes in pre-clinical models and clinical translation remains elusive [7,9,18,19,20,21]. In this review, we have focussed on the specific targeting of matrisome components to try to unravel causative and correlative relationships. Examining individual matrix components often reveals strikingly complex extracellular signalling roles, which challenges us to reconsider the importance of the ECM in the processing of information. The most promising strategies for cancer treatment appear to work through modulation of the immune landscape, and successful targeting of the stroma to improve outcome and therapy response remains an exciting prospect.

## 2. Common Matrix Signatures Evolve to Support Invasion and Metastasis

While the matrices in distinct tissues each have their own characteristic composition and biomechanical properties, commonalities also exist: the primary matrix component in most tissues is fibrillar type 1 collagen; most parenchymal tissues and early stage tumours are surrounded by basement membranes, a specialised type of barrier matrix rich in collagen IV and laminins; and the matrix provides a scaffold for the parenchyma and vascular trees providing access for nutrients and immune cells and egress for waste. All solid tumours must subvert these physiological properties to flourish. The rapid advance of omics technologies has allowed us to catalogue temporal and spatial changes to the matrisome during tumour progression for diverse cancer types uncovering both commonalities and tumour-specific mechanisms.

In solid cancers, the matrix co-evolves with the malignant component to support tumour growth, evade immune surveillance, and promote disease progression [22]. Many studies have revealed conserved matrisome features and signatures associated with poor outcome and metastasis (Table 1) [7,8,23,24,25]. Moffit et al. used an in silico approach to separate stromal and epithelial signatures from bulk transcriptomic data and used this to define signatures of the activated vs. normal pancreatic cancer stroma; interestingly, the prognostic power of the activated stromal signature was dependent on the epithelial subtype, predicting worse outcome in basal-like tumours [26]. Most identified genes were matrix-related, including many collagens and FN1. Pearce et al. employed a multi-omics approach to examine features associated with metastatic progression in ovarian cancer [8]. They defined a transcriptomic Matrix Index (MI) scoring system, based on the 22 genes most significantly associated with disease severity and tissue stiffness. Importantly, the MI is highly prognostic, not only for ovarian cancer but also for many epithelial and mesenchymal solid tumours. The MI correlated robustly with regulatory T-cell content (FOXP3+), though a relationship with CD8+ T-cells was less clear. Many genes overlap between the MI and Moffitt signatures, despite distinct tissue origins, including COL1A1, FN1, VCAN, and COMP.

In breast cancer, matrisome signatures associated with metastasis have been documented, highlighting FN1, Tenascin C, and lysyl oxidases as poor prognostic indicators [23]. Collagen I again features in this prometastatic signature, alongside COLV and COLVI; importantly, COLVI has also been associated with a functional role in TNBC metastasis in a cell-derived matrix model [27], while COL1A1 or FN1 alone can also drive an aggressive breast cancer phenotype in culture models [28]. Indeed, both COL1A1 and FN1 have been identified as individual prognostic biomarkers in a variety of cancers [29,30,31,32,33,34,35,36]. Taking a proteomic approach to map the evolving matrisome in breast cancer, Papanicolaou et al. identified and characterised a role for collagen XII as a metastasis-related component, though, interestingly, this was functionally linked to regulation of collagen I fibril organisation [37].

A picture emerges where the tumour-associated matrix promotes an immune-evasive, pro-metastatic environment, often through modulation of common and abundant matrix components, including collagen I, FN1, and hyaluronic acid (HA). Organisation, alignment, crosslinking, and degradation all play important roles, which offer numerous therapeutic opportunities.

## 3. Matrix Signatures Associated with Immune Evasion and Immune Checkpoint Response

Historically, tumour matrix studies have focussed on invasion and metastasis, whereas recent emphasis has turned to immune evasion (Table 1). Significant effort is now being made to understand the role of the tumour microenvironment (TME) in the regulation of anti-tumour immunity and ICB response. Taking a pan-cancer approach, Chakravarthy et al. characterised ECM changes between normal and malignant tissue across multiple solid tumour TCGA datasets and defined a conserved gene expression signature associated with malignancy [25]; notably, the cancer-associated ECM signature (C-ECM) is not only associated with disease prognosis but also predicts immune evasion and immune-checkpoint blockade (ICB) failure. Many of the ECM genes identified are TGFβ-responsive targets in cancer-associated fibroblasts (CAFs), and these data concur with a broad range of studies implicating the TGFβ-activated stroma with poor ICB response [38,39,40,41]. The intimate link between the matrisome and immune content implies that the direct targeting of specific matrisome components may provide benefit to augment immunotherapy and chemotherapy response [6].

In breast cancer, an early stromal gene expression study by Finak et al. also highlighted the critical contribution of the stroma–immune relationship to breast cancer prognosis [42]. A clinically useful example is provided by the colorectal cancer (CRC) immunoscore, largely based on CD3 and CD8 T-cell content. The immunoscore is a powerful predictor of both prognosis and ICB response in MSI-positive CRC tumours [43,44,45]; immunotherapy is now the frontline therapy for the management of MSI-positive metastatic CRC, where biomarkers for response are critical [46]. Similarly in melanoma, where immunotherapy is offering huge clinical benefit, signatures associated with T-cell exclusion are powerful predictors of ICB response [47].

While T-cell content is a powerful predictor in some cancer settings such as CRC, the relationship between CD8+ T-cells and ICB response is clearly complex, as many factors can govern the ability of T-effector cells to target tumour cells, including neo-antigen load, PD-L1 expression, T-cell distribution, and the presence of immune-suppressive Tregs, γδ T cells, and myeloid suppressor cells [48,49,50,51]. To integrate these factors, the Tumour Immune Dysfunction and Exclusion study (TIDE) defined a scoring system which considers many critical factors governing anti-tumour immune responses, including T-cell exclusion, T-cell exhaustion, immune cell repertoire, and stromal signatures [52]. Examining melanoma trial results, the study concludes that high CD8+ T-cell content is an effective indicator of ICB response only when associated with a low TGFβ signature [52]. The TME risk signatures developed for lung [53] and HCC [54], as well as the immune checkpoint inhibitor scoring system (IMS) for bladder cancer [55], provide additional informative examples, defining TME features associated with anti-tumour immunity and ICB response. In each of these signatures or scoring systems, matrisome components are abundant, with commonalities regarding TGFβ signatures, collagen regulation, and immune content.

**Table 1 cancers-16-01847-t001:** Stromal signatures associated with prognosis or therapy response.

Name	Derived	Application	Cancer Focus	Number of Genes
Dominguez et al. [39]	Marker genes for a CAF subpopulation	Predicts anti-PD-L1 therapy response	Pancreatic ductal adenocarcinoma	12 upregulated genes
Pearce et al. [8]	Matrix genes associated with disease score and tissue modulus	Predicts poor prognosis	High-grade serous ovarian cancer	6 upregulated genes, 16 downregulated genes
Moffitt et al. [26]	Differential expression of stromal genes between CAFs and tumour cell lines	Predicts poor prognosis	Pancreatic ductal adenocarcinoma	48 upregulated genes
Murray et al. [24]	Transcriptome associated with PKN2 knockout	Predicts poor prognosis	Pancreatic ductal adenocarcinoma	11 upregulated genes
Öhlund et al. [56]	Transcriptome associated with iCAF subtype	CAF subtype discrimination	Pancreatic ductal adenocarcinoma	200 upregulated genes, 200 downregulated genes
Jiang et al. [52]	Transcriptome associated with T-cell exclusion and dysfunction	Predicts immune checkpoint inhibitor response	Multiple	770 genes
Yan et al. [53,54]	TME signatures associated with anti-tumour immunity	Predicts poor prognosis and immune checkpoint inhibitor response	Hepatocellular carcinoma	Four downregulated genes
Lin et al. [55]	Gene expression of 27 survival-related immune signatures	Predicts good prognosis and immune checkpoint inhibitor response	Gastric cancer	463 upregulated genes
Brechbuhl et al. [23]	Proteome associated with CD146–CAFs	Associated with increased risk of metastasis	Breast cancer	12 upregulated genes
Wang et al. [57]	Regression in TCGA ESTIMATE scores	Predicts poor prognosis	Gastric cancer	Three upregulated genes, one downregulated gene
Jia et al. [58]	Regression in TCGA ESTIMATE scores	Predicts poor prognosis	Colon adenocarcinoma	Three upregulated genes
Yue et al. [59]	Transcriptome associated with TCGA lymphovascular space invasion	Predicts poor prognosis	Serous ovarian cancer	Eight upregulated genes
Isella et al. [60]	CAF-expressing genes associated with stem/serrated/mesenchymal (SSM) transcriptional subtype	Predicts poor prognosis	Colorectal adenocarcinoma	130 upregulated genes
Farmer et al. [61]	Genes co-expressed with decorin	Predicts poor neoadjuvant chemotherapy response	ER-negative breast cancer	50 upregulated genes
Boersma et al. [62]	Stromal genes differentially expressed between inflammatory and non-inflammatory breast cancer	High inflammation	Inflammatory breast cancer	2 upregulated genes, 20 downregulated genes
Strell et al. [63]	Differential expression on platelet-derived growth factor (PDGF)–activated human fibroblasts	Predicts poor prognosis	Early breast cancer	55 upregulated genes, 58 downregulated genes
Casey et al. [64]	Differential expression between invasive cancer stroma and normal stroma	Associated with invasion	Breast-invasive carcinoma	Nine upregulated genes, five downregulated genes
Winslow et al. [65]	Differential expression between LCM stromal TN breast cancer tumours. Correlating gene changes with TCGA	Predicts poor prognosis	Breast-invasive carcinoma	53 upregulated genes, 26 downregulated genes

Taking several matrix signatures together, we also examined the overlap between genes to identify commonalities; 16 genes appeared in at least three independent signatures, including COLA1, COL11A1, VCAN, POSTN, and FN1 (Figure 1). In the next sections, we will first consider how distinct CAF subtypes are associated with both matrisome signatures and tumour immunophenotype. Then, we will address the degree to which the matrisome can dictate, rather than correlate with, immunophenotypes by considering both CAF-directed and individual-matrisome-component-directed interventions; we will focus on matrisome genes common to multiple predictive signatures (Figure 1).

## 4. Cancer-Associated Fibroblasts Are Principal Contributors to the Matrisome

CAFs are considered the dominant source of the TGFβ-stimulated pro-tumourigenic matrisome and many recent spatial and single-cell transcriptomic studies have characterized distinct CAF sub-populations, which evolve with advancing malignancy. A seminal study by the Tuveson lab identified a major dichotomisation of CAFs in PDAC into myofibroblast CAFs (myCAFs) and inflammatory CAFs (iCAFs), since confirmed in numerous scRNA-seq studies across many solid cancers [39,56,57,58,59,60,61,62,63,64,65]. Typically, myCAFs are driven through activation of TGFβ signalling, while iCAFs are reported to be driven by inflammatory cytokines [58]. Differential spatial locations within tumours and reciprocal interaction between the TGFβ and IL1 signalling pathways appear to drive distinct lineage trajectories [7,62]. Numerous additional CAF subtypes have since been described, including those defined by high matrix deposition (ecm-CAFs [64] and matrix-CAFs [57]). The cataloguing of fibroblast heterogeneity in solid tumours has supported the idea of tumour-promoting and tumour-supressing CAF populations, but it has also revealed the complexity of targeting these cells due to overlapping function and a lack of discriminatory biomarkers and drug targets [7,62].

### 4.1. Targeting CAF Subsets to Modify Tumour Biology

Approaches to selectively target CAF subtypes have yielded mixed results. In 2014, a group of important studies, which variously suppressed myofibroblast populations in PDAC, suggested that CAFs may play a tumour-suppressive role [19,20,21]. Recent studies have revealed a more nuanced picture with distinct CAF subsets playing opposing roles in tumours. Exciting recent work from the Turley lab elegantly mapped the evolution of CAF subsets as PDAC progresses in mouse models to define a TGFβ-programmed LRRC15+ CAF population associated with advanced disease [39,40]. Importantly, this CAF signature was identified in many solid tumour types and associated with poor ICB response in trial data. Ablation of LRRC15+ CAFs in mouse models was also able to enhance anti-tumour immunity and ICB response [40]. Interestingly, this study also revealed that both myCAF and iCAF lineages can express abundant collagen I and II, suggesting a more complex relationship between myCAF subtype and fibrosis. Hutton et al. also identified distinct CAF lineages in PDAC with distinct CD105-positive CAF-promoting tumour growth, while CD105-negative CAFs support robust anti-tumour immunity [41]. Interestingly, in the 2014 Ozdemir et al. study in which α-SMA CAF ablation drives more aggressive PDAC growth, tumours were also rendered more sensitive to anti-CTLA4 immunotherapy, perhaps indicating distinct tumour-restraining and immune-suppressive roles for overlapping CAF populations. TGFβ-driven CAF subsets supporting immune evasion have also been defined in mouse models of breast and other cancers [7,38,59,66]. In TNBC, distinct CAF lineages have also been associated with CD8+ T-cell exclusion or functional suppression [60]. The causal, prognostic, predictive association between TGFβ-induced stromal signatures and the suppression of anti-tumour immunity and ICB response has now been robustly established [7,38].

Powerful spatial transcriptomic studies have also demonstrated that individual glands within tumours can harbour highly localised relationships between cancer cells, CAFs, and immune cells [67,68]; advanced tumours can contain many distinct cancer-subtypes and localised TME interactions, which must be taken into account when considering therapy response and resistance [69]. In the next section, rather than focussing on targeting the complex and localised interaction between CAFs and tumour cells, we wish to explore interventions which specifically target matrisome components.

### 4.2. Targeting TGFβ and CAF Activation

Many TGFβ- and CAF-targeting strategies have been shown to suppress stromal activation and tumour fibrosis, but interpretation is often complex due to multimodal mechanisms of action. This is exemplified by studies targeting the Hedgehog pathway where both positive and negative impacts on tumour progression and therapy response have been reported [9,20,61]. Co-administration of TGFβ-blocking antibodies or inhibitors with anti-PD-L1/PD1 has proven successful in mouse models [38,66]. Indeed, inhibitors of TGFβ signalling, including Galunisertib, NIS793, and LY364947, have all reached clinical testing for a variety of solid tumours, though concerns remain regarding tumour-suppressor roles in cancer cells for this signalling axis [7,70,71,72,73]. The angiotensin antagonist Losartan can suppress TGFβ release and fibrosis in a variety of pathological settings, including pancreatic and other solid tumours. In fibrotic breast and pancreatic cancer models, Losartan has been shown to reduce CAF-derived collagen and HA to enhance perfusion and potentiate chemo- and immunotherapy responses [74,75,76], and this has translated to promising clinical trial results [77,78]. The anti-fibrotic drug Pirfenidone also limits fibrosis by targeting TGFβ signalling and CAFs in both breast and pancreatic cancer models [79,80,81,82]. Numerous activating signalling axes in CAFs have also been the subject of significant studies, including Hedgehog signalling, Vitamin A and D pathways, FAK, and PKN2 [7,9,24,83,84,85]. Clinical trials targeting CAFs and TGFβ, while largely disappointing, have shown some promise in subsets of patients (reviewed [7,86]). The multimodal mechanisms of action of these important interventions lies beyond the scope of this review.

## 5. Direct Matrix—Interventions

While it has long been established that ECM signatures correlate with tumour progression and therapy responses, therapeutic targeting of specific matrisome components has proved difficult to enact [87]. Is this because ECM signatures are correlative but not causative with advancing malignancy, or can the matrix be used to dictate tumour phenotypes? Here, we will focus primarily on the direct targeting of matrisome components, rather than the targeting of TME cells to remodel the stroma. We will address efforts to both degrade the matrisome to support therapy and efforts to protect the matrix from degradation to prevent invasion and metastasis. The opposing need to prevent cancer cell egress while promoting the ingress of anti-tumour immune populations presents a unique dilemma when targeting the matrisome, which may underlie poor clinical success. The onus is on identifying how altering the matrisome composition can dictate beneficial tumour phenotypes, while maintaining tumour-suppressive functions.

### 5.1. Is Tumour Fibrosis Good or Bad?

Among the predominant fibrous proteins in the ECM are collagens, elastin, fibronectin, and laminins. Laminins are largely associated with basement membranes, playing key roles in tissue organisation, tumour invasion, and metastasis, as reviewed elsewhere [88,89,90]. The interstitial matrix is largely composed of collagen fibres, crosslinked with elastin fibres, and interspersed with varying levels of HA and proteoglycans (Figure 2). The general impact of high levels of desmoplasia on tumour prognosis is complex, but it is most often associated with poor outcomes in advanced cancer [91,92,93,94] (Figure 2); whether fibrosis is correlative with aggressive disease or causative remains an area of significant debate. Here, we will examine evidence for pro- and anti-tumour roles for central ECM components.

*Collagen*. The most abundant protein component of the TME, and indeed the human body, is type 1 collagen, and collagen-rich tumour fibrosis has long been considered as a barrier to therapy, particularly in fibrotic tumours such as PDAC [18,75,95,96]. Furthermore, aligned collagen has been identified as a functional promoter of invasion and poor-prognosis biomarker in breast cancer [97,98,99,100]. Perhaps surprisingly, conditional deletion of Col1a1 from α-SMA-positive CAFs (α-SMA-Cre/Col1a1^fl/fl^) crossed with a *Kras^G12D/+^*; *Trp53^frt/frt^*; *Pdx1-Flp (KPPF)* mouse model of pancreatic cancer resulted in reduced fibrosis but more aggressive tumours, lower survival, and myeloid suppression of CD8+ T-cells (Figure 2B) [101]. In various additional mouse models of PDAC, other myofibroblast CAF-targeting approaches have also led to reduced tumour fibrosis, often resulting in more aggressive and immune-suppressive tumours [19,20,21,24,102]. A tumour-restraining role for CAF-derived type 1 collagen has also been reported for growth of PDAC and CRC liver metastases; collagen 1 was shown to physically restrain tumours using a hepatic stellate cell (HSC)-specific Col1a1^fl/fl^ conditional knockout model [102]. The emerging consensus thus suggests a tumour-restraining role for CAF-derived type 1 collagen in multiple mouse models [101,102].

In the same liver metastasis study, Bhattacharjee et al. additionally showed that CAF-derived HA and HGF were pro-tumourigenic and that depletion of α-SMA-positive HS-derived CAFs resulted in slower growth and enhanced survival [102]. This reveals tumour-promoting and -suppressing roles for the same CAF population to highlight the challenges associated with CAF ablation strategies [102]. Adding further nuance, targeted deletion of Col1a1 specifically from the pancreatic cancer cell compartment in KPPC; Col1^pdxKO^ mice (*LSL-Kras^G12D/+^*; *Trp53^loxP/loxP^*; *Pdx1-CreCol1a1^loxP/loxP^*) increased mouse survival by suppressing the formation of oncogenic Col1a1 homotrimers [103]; moreover, Col1 targeting in this context enhances T-cell infiltration and checkpoint response while also impacting the tumour microbiome. Tumour-cell-derived Col1a1 has also been reported as a metastatic driver in breast and liver cancer models [32,97,104]. Cell-type-specific conditional targeting of Col1a1 in mouse models of breast and other solid cancers are yet to be reported. Thus, while collagen 1 genes feature in a broad number of poor-prognosis signatures, they currently do not readily represent a tractable therapeutic target.

While type-1 collagen is the most abundant collagen, at least 28 collagen types have been described with diverse functions in cancer [105]. Some consensus emerges for some collagen types, including type 6 and 11. Col11A1 is a key upregulated component of the Matrix Index, which predicts metastasis across many tumour types, including ovarian, pancreatic, and breast cancers [8]. Col11a1+ CAFs, which co-express LRRC15, are also associated with tumour progression in scRNA-seq and tissue-staining studies [106,107]. Other studies have identified COL11A1 as a functional driver of cancer progression and prognostic biomarker [108,109,110]. A similar body of the literature is emerging supporting pro-tumourigenic functions for COLVI in breast, ovarian [111], pancreatic [112], lung [113], and brain [114] cancers. Clinical intervention here is yet to be explored.

In summary, while fibrosis is a common feature of many aggressive treatment-resistant tumour, and a known activator of oncogenic mechanotransduction signalling axes, relief of collagen-rich fibrosis alone may promote rather than improve outcome.

*Fibronectin*. PhysiologicallyC, fibronectin is produced by myofibroblasts at sites of injury; as cancers can be considered as “*wounds that never heal*”, FN is generally upregulated in the tumour ECM, where high expression is often associated with invasive disease and poor prognosis. As FN is upregulated in many cancers, it has been exploited primarily in cancer imaging and for the targeted delivery of therapies; numerous antibodies, peptides, and modified ligands have been used to deliver isotopes, chemotherapeutics, and immunomodulators to tumours in pre-clinical models (reviewed [115,116]). Fibronectin has also emerged as an attractive target for guiding CAR-T therapy, particularly focused on cancer-specific splice variants [117,118,119,120,121]. Mechanistically, fibronectin can promote both growth and invasion through engagement of α5β1 and αv-class integrins, and this provides therapeutic opportunities [122,123,124,125]. Several monoclonal antibodies (mAbs) preventing integrin interaction with FN, including those against αvβ6 and α5β1, have shown excellent pre-clinical promise in breast and glioma models, respectively [126,127], highlighting the value of this axis. Surprisingly, few studies have been published examining the conditional deletion of FN in pre-clinical tumour models, particularly given the well-documented roles in promoting invasive disease. Interestingly, conditional deletion of circulating plasma fibronectin, through targeted deletion in hepatocytes, reduces MDA-MB-231 bone metastatic colonisation in nude mice [128]. Suppression of FN expression in MDA-MB-231 has also been shown to reduce bone metastases [129]. Suppression of FN expression by microRNAs also limits glioma progression in a mouse model, adding further evidence of therapeutic potential [89]. Conditional deletion from CAFs—the predominant source of fibronectin in many tumours—remains to be reported. The importance of matrix proteases in the degradation of FN will be addressed later in this review.

### 5.2. Degrading Hyaluronic Acid

A direct ECM depletion approach which has gained more traction has been the targeting of hyaluronic acid (HA, Hyaluronan), an abundant glycosaminoglycan ECM component in solid tumours, which can be a dominant contributor to tumour stiffness, hydration, and interstitial pressure [18,130,131]. HA and its two main receptors, CD44 and RHAMM, have been implicated in inflammation in both physiological settings and cancer [132,133]. Dysregulation of HA synthase enzymes (HAS1-3) has also been reported for many cancers including pancreatic, breast, and prostate cancer [134,135,136,137]. Numerous studies have implicated high levels of HA as a prognostic indicator and functional driver of poor outcome in solid tumours (Figure 2C) [138,139,140,141,142]. Importantly, HA can be degraded by hyaluronidases, and the use of exogenous hyaluronidases as drugs to augment cancer therapy has a long history, although early translation was limited by poor drug characteristics [143,144]. PEGylated human hyaluronidase (PEGPH20) was developed to improve pharmacokinetics, showing activity in high-HA preclinical prostate tumour xenografts [145]. PEGPH20 also sensitised pre-clinical PDAC models to chemo-, radio-, and immunotherapy approaches [10,18,145,146,147]. Increased chemotherapy uptake and response in preclinical ovarian cancer mouse models have also been reported [148]. Phase II and phase III trials combining PEGPH20 with chemotherapy to treat advanced PDAC have followed; although failing to meet primary endpoints, some response improvement was observed in HA-rich tumours [149,150,151,152]. Additional combinatorial trials are underway for PDAC and other solid cancers [153]; stratifying for tumours with high-HA content is likely to be critical to improve trial success [145,149,150].

In breast cancer, PEGPH20 has been shown to improve HER2-targeting antibody therapy response [154] and anti-PDL1antibody uptake and therapy responses [155] in mouse xenograft models. In addition to facilitating therapy uptake, depletion of HA may also limit inflammatory HA-CD44 signalling. In breast cancer xenograft models, depletion of HA or loss of CD44 were both shown to reduce CCL2 production in vitro and loss of CD44 impaired macrophage recruitment and tumour induction in vivo [156]. A phase II metastatic breast cancer trial combining Eribulin with PEGPH20 was terminated (NCT02753595) and to date PEGPH20 trial results in breast cancer are yet to be reported. Although translation has been challenging, the positive impact of PEGPH20 on immune infiltration, drug perfusion, and pre-clinical tumour model drug responses provides encouragement that modifying the biophysical properties of the stroma can enhance clinical impact.

## 6. Matrix Crosslinking

In addition to changes in composition, the stiffness of the ECM is critically regulated by crosslinking, with the lysyl-oxidase (LOX) family of collagen crosslinkers playing a dominant role. LOX and LOX-like (LOXL) enzymes catalyse the crosslinking of collagen and elastin, and this promotes matrix stiffening in pathological and malignant settings. The targeting of LOX enzymes, using function-blocking mAbs, has long shown promise in normalising a pathologically stiff ECM in mouse xenografts and fibrosis models [157,158]; the targeting of LOX can reduce stiffness, desmoplasia, invasion, and metastasis. High levels of LOX family members have also been functionally implicated in disease progression and correlated with poor outcome in many fibrotic tumour types, including pancreatic, breast, and lung cancer [159,160,161,162,163,164,165].

In a pre-clinical MDA-MB-231 xenograft model of TNBC, mAb targeting of LOX was shown to limit chemotherapy resistance by improving drug penetration and through regulation of FN1 and ITGAV expression. Upregulation of LOX, FN1, and ITGAV is also seen in relapsed TNBC patients, providing data to support translation [166]. Rossow et al. further demonstrated that LOX enzymes limit drug access in a range of solid tumour xenograft models, using over-expression studies and the pan-LOX small-molecule inhibitor bAPN [167]. Nanovesicle delivery of anti-LOX mAbs has also been validated in MDA-MB-231 xenografts [168]. Small-molecule inhibitors of LOXL2 have also been developed, with the potential to reduce MDA-MB-231 xenograft metastasis and tissue fibrosis [169,170,171].

As with other interventions to supress stromal fibrosis, conflicting results have emerged in pancreatic cancer models. LOX targeting with mAbs in KPC mice potentiated gemcitabine responses, reduced metastasis, and extended survival; mechanistically, these enhanced responses were associated with enhanced myeloid cell infiltration, reduced fibrillar collagen, and improved vasculature, rather than enhanced levels of gemcitabine delivery [172]. Genetic deletion or cancer cell overexpression of LOXL2 in KPC and KC autochthonous models also support a tumour- and metastasis-promoting role [173]. In contrast, LOX targeting in a syngeneic orthotopic PDAC model resulted in more aggressive disease [174], in line with distinct CAF depletion and collagen-targeting strategies [19,20]. bAPN has, however, been shown to facilitate drug perfusion in PDAC orthotopic xenografts in nude mice [175]. A clinical trial combining the LOXL2-targeting mAb Simtuzumab with gemcitabine did not, however, improve outcome, though this may reflect advanced disease stage and a lack of patient stratification (NCT01472198 [176]). Simtuzumab has also been explored clinically for treatment of pulmonary and liver fibrosis, though results remain disappointing [177,178,179,180].

## 7. Matrix Proteases

Metalloproteinases, including MMPs, ADAMs, and ADAMTSs, constitute large families of proteases which can degrade various components of the ECM (ECM). They have been broadly implicated as facilitators of invasion and metastasis, by degrading the basement membrane and stromal desmoplastic matrix, and as such have been the subject of extensive drug discovery programs. To date, clinical translation has been largely unsuccessful, which has largely been attributed to a lack of inhibitor specificity, combined with tumour-suppressive functions and the normal physiological roles of related family members. Despite the disappointing results from numerous clinical trials, hope remains that protease-targeting drugs can be of use in cancer (clinical trials reviewed by Coussens et al. [181] and Cathcart et al. [182]). Roles for specific MMPs are nuanced and show both disease- and context-dependent functions. Here, we will cover some recent examples where antagonistic behaviours of close protease family members are likely to confound therapeutic targeting with non-selective small molecules. Alternative strategies, such as mAb or gene-suppression approaches, may present alternative isoform-specific strategies.

Regarding matrix degradation and remodelling, distinct matrix proteases can either support or impede tumour invasion. While proteases may facilitate invasion, distinct family members which degrade or remodel a pathogenic pro-invasive matrisome can also act as tumour suppressors. DCIS provides some excellent illustrative examples. In DCIS, luminal tumour cells proliferate within breast ducts but are restrained from invading the surrounding stroma by myoepithelial cells and basement membrane surrounding the ducts. In progressive disease, disruption of the basement membrane has been associated with upregulation of pro-invasive MMPs (e.g., MMP9 and MMP13) by stromal cells, including myoepithelial cells and CAFs (Figure 3A) [183,184,185,186,187]. In many DCIS cases, however, progression to invasive carcinoma does not occur. Discriminating between indolent disease and those cases likely to progress to invasive carcinoma remains an unmet clinical demand. Accumulation of fibronectin has been associated with disease progression and poor outcome, potentially by promoting myoepithelial-led invasion [122,188,189,190]. A recent study demonstrated that ADAMTS3, produced by myoepithelial cells, can restrict invasion by degrading fibronectin in a DCIS model [191]. Loss of ADAMTS3 supports the accumulation of fibronectin to promote an integrin-α5β1-directed myoepithelial-led invasion (Figure 3A). Similarly, MMP8, which is lost in invasive disease, has been shown to act as a tumour suppressor through promotion of adhesion and suppression of MMP-9 function [192]. Sparing proteases responsible for fibronectin degradation (e.g., ADAMTS3) may thus be desirable in a therapeutic setting. The challenge of developing inhibitors to discriminate between closely related proteases with antagonistic functions can help explain the poor performance of broad MMP inhibitors in clinical trials.

In pancreatic cancer, the desmoplastic stroma is considered a barrier to therapy, but numerous recent studies also suggest a robust tumour-suppressive function, where suppression of fibrosis has been associated with invasive and metastatic disease. A recent report by Carter et al. highlights antagonistic roles or the closely related collagenases, ADAMTS2 and ADAMTS14 in the promotion of pancreatic cancer invasion (Figure 3B) [193]. Suppression of ADAMTS2 was found to limit pancreatic stellate cell (PSC) activation, while suppression of ADAMTS14 promoted PSC myofibroblast differentiation and cancer cell invasion in heterotypic 3D models. Mechanistically, these two proteases differentially target the distinct matrisome substrates, serpinE2 and fibulin 2, to regulate TGFβ availability. Of note, antagonistic regulation of Fibulin 2 cleavage by ADAMTS5 and ADAMTS12 has been previously reported to promote invasive behaviour in breast cancer cells, potentially through production of bioactive Fibulin 2 proteolytic products [194]. These examples illustrate nuanced, substrate-specific signalling capability for this family of enzymes, akin to behaviours observed in intracellular signalling cascades.

## 8. Matrix Protease Regulation of the Immune Landscape

While protease modulation of the TME may be pro-invasive in many scenarios, it is also clear that MMPs and other matrix degrading proteases can also play tumour-suppressive roles, not least through their importance in enabling immune cell access [195]. While the matrix can act as a protective barrier preventing tumour cell escape, impeding immune cell access also provides a safe haven for cancer cells. Ager et al. demonstrated that anti-MMP14 mAb DX-2400 alone was able to enhance anti-tumour immunity to impede tumour growth and enhance radiotherapy response in 4T1 nude mouse xenografts and syngeneic E0771 immune-competent models [196]. Increased M1 macrophage infiltration, with downregulation of TGFβ and SMAD pathways, suggests remodelling of the immune-suppressive stroma. Similar results were reported for a second MMP-14-targeting antibody using MDA-MB-231 xenografts and the syngeneic 4T1/BALB-c metastasis model [197]. Targeting MMP-9 has also showed promise for enhancing immunity. Anti-MMP9 was found to slow growth of HER2+ mouse orthotopic breast tumours, while combination with anti-PDL1 enhanced effector T-cell infiltration [198]. Intriguingly, the authors demonstrated that MMP9 was capable of directly cleaving and inactivating T-cell chemokines CXCL9/10/11, and that inhibition with anti-MMP9 could enhance CXCL10 in vivo through this mechanism (Figure 3C) [198]. This again highlights substrate-specific roles for matrix proteases, beyond remodelling the ECM. Owyonng et al. separately confirmed that anti-MMP-9 can enhance CD8+ T-cell content in the MMTV-PyMT breast cancer model [199]. Excitingly, in mouse lung and melanoma models, SB-3CT, a small-molecule MMP2/9 inhibitor, has been shown to enhance PD-1 or CTLA-4 blockade responses by promoting anti-tumour immunity [200].

The ADAMTS proteases have also been associated with immune modulation and in particular with inflammation, although cancer studies remain somewhat limited [201]. ADAMTS1 acts as a tumour promoter in the genetic MMTV-PyMT model, with ADAMTS1 knockout mice having reduced tumour burden, metastasis, and prolonged survival [202], associated with enhanced leukocyte infiltration and cytotoxic immune signatures. Tumour-promoting roles for ADAMTS1 have also been reported in melanoma, and ADAMTS1 deficiency can promote a pro-inflammatory landscape in melanoma [203,204]. Similarly, ADAMTS4 has been associated with high-macrophage and tumour-promotion content in a subcutaneous CRC xenograft model [205]. In contrast, a tumour-suppressor role has also been reported for ADAMTS1 in breast cancer, related primarily in this study to the regulation of tumour vasculature [206].

Among the substrates of ADAMTS proteases, the proteoglycan versican has shown most promise as an immune modulator [207,208]. Interestingly, ADAMTS-derived cleavage fragments of versican (e.g., versikine [209]) have been shown to modulate key aspects of the tumour immune landscape, including modulation of DCs and CD8+ T-cell recruitment in myeloma, lung cancer, and CRC (Figure 3D) [210,211,212,213]. Recent work shows that versikine enhances DC abundance and activity in a Lewis Lung Carcinoma (LLC) model [214]. In CRC, VCAN proteolysis and versikine levels are associated with robust CD8+ T-cell infiltration, also likely driven through modulation of DC numbers and activation status [210]. Furthermore, versikine levels have been correlated with ICB (pembrolizumab) response in a Phase 1 clinical trial for metastatic CRC (NCT02837263). The action of versikine, and other ‘matrikine’ cleavage products, once again reveals complex regulator roles for specific protease–ECM interactions. Matrikine cleavage products with biological function in cancer have been reported for many matrix components including elastin [215,216,217,218]. Matrikine production during remodelling of the ECM is an area of matrix biology which is sure to gather pace with the application of novel degradomics technologies [219,220]. Examples provided in this review suggest complex signalling networks will emerge in the extracellular space, perhaps under-appreciated due to our historic reliance on limited tissue culture models. Therapeutically, the presence of such extra-cellular signalling networks opens numerous opportunities for the development of biological therapies (e.g., mAbs).

Together, these findings suggest isoform-specific protease inhibition may be of value in combination with ICB in a variety of solid tumour settings. Proteases can act through novel immunomodulatory mechanisms beyond depletion of matrix components, including the generation and degradation of immunostimulatory chemokines and matrikines.

## 9. Matrisome Interventions to Sculpt the Immune Landscape

A recurrent theme in the majority of promising therapeutic interventions targeting the tumour matrisome is promotion of anti-tumour immunity or sensitisation to ICB inhibitors. Attempts to degrade or suppress the collagen-rich matrix have largely failed, due to induction of an immune-suppressive, myeloid-rich TME, although in some cases this does not tell the whole story. Multiple mouse model studies on PDAC suggest that more aggressive inflammatory tumours, promoted by stromal suppression, may be sensitised to ICB inhibitors. As an example, ablation of α-SMA CAFs in a genetic PDAC model (PKT: *Ptf1a*^cre/+^; *LSL-Kras*^G12D/+^; *Tgfbr2*^flox/flox^) suppressed fibrosis and accelerated tumour growth, but it also was sensitised to anti-CTLA-4 therapy, associated with increased inflammation. Similarly, targeting the Hedgehog pathway to limit stromal activation has been shown to promote aggressive inflammatory pancreatic tumour growth [20]; Hedgehog inhibitors, nonetheless, also show promise in combination with chemotherapy or immunotherapy regimens in pancreatic [6], ovarian [9,221], and other solid cancers. These approaches are yet to make an impact clinically (reviewed [7,86,222]). Likewise, preclinical successes with hyaluronidase PEGPH20 have been associated with promotion of anti-tumour immunity and ICB responses in pancreatic and breast models [147,155].

While numerous matrisome signatures have been shown to correlate with prognosis and ICB response, increasingly, the evidence supports direct causative functions. The ability of the matrisome to dictate immune cell phenotypes has elegantly been demonstrated using decellularised-tumour-derived matrices. Puttock [223] et al. adopted an ex vivo approach using a decellularised ECM from mouse ovarian cancer metastases to examine immune modulation. Monocytes cultured on a decellularised ECM were found to differentiate into immunoregulatory M0 phenotype macrophages. Furthermore, these matrisome-educated macrophages could promote T-cell proliferation and expression of T-cell activation markers [223].

Versican appears in several poor outcome matrix signatures, including the Matrix Index and C-ECM [8,25]. Versican interacts with many of the ECM components addressed in this review, including HA, FN1, and MMPs, and several studies have delineated roles in the modulation of myeloid cells and inflammation [207,224]. The versikine studies described above indicate a positive functional impact on the ability of DCs to prime adaptive immunity and promote ICB response. Versican can also directly activate macrophages to stimulate TNF-α secretion and lung tumour growth in the LCC model [225]. Versican modulates TAM phenotype in mesothelioma, where high levels correlate with poor outcome [226]. Mechanistically, versican-deficient tumours in mice are less aggressive and have fewer macrophages and neutrophils. Importantly, versican-deficient mesothelioma cells polarised co-cultured macrophages to an M1 phenotype, suggesting VCAN targeting may promote a more favourable immune landscape. Versican may also act as a direct regulator of T-cell trafficking, likely in complex with HA [227]. Indeed, versican predominantly binds to HA in tumours and degradation of the HA matrix with PEGPH20 will act in part by altering versican and versikine distribution, with repercussions on myeloid and lymphocyte recruitment [132,227,228].

## 10. Summary

The concept that tissue composition, and not just cancer cell properties, is critical in deciding cancer outcome was first proposed in 1889 when Paget described his seed and soil theory of metastasis [229,230,231]. The matrix is now understood to regulate all stages and hallmarks of cancer, from initiation, growth, and metastasis to disease recurrence and therapy resistance. With the emergence of diverse omics technologies, our ability to map the evolution of the matrisome in time and space, as cancers progress, is uncovering a multitude of mechanisms and therapeutic opportunities. Most matrix-directed therapeutic approaches have focussed on attempting to limit cancer spread or improve therapeutic responses; in many cases, these two aims are at odds with one another. The same matrix which limits drug or immune cell access can also suppress tumourigenesis by restraining cancer growth and spread. Most successes in pre-clinical models from diverse cancer types have been associated with improving anti-tumour immunity, although mechanisms are unsurprisingly diverse. Regulating the matrisome to sculpt the immune landscape has enormous promise in combination therapies. Significant challenges remain with clinical translation, not least with spatial heterogeneity within tumours and between patients. Better stratification, examining both tumour and stroma, will help unlock the promise of matrisome interventions, particularly to reawaken anti-tumour immunity.

## Figures and Tables

**Figure 1 cancers-16-01847-f001:**
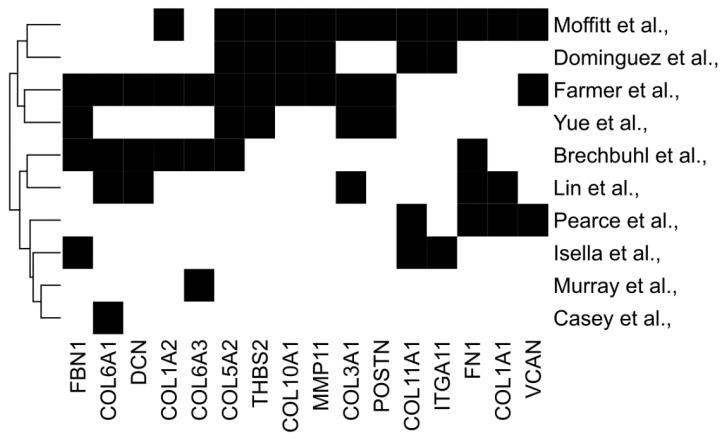
Commonalities between diverse predictive matrix signatures. Common genes from several stromal signatures associated with prognosis or therapy response. Genes selected participate in three or more stromal signatures.

**Figure 2 cancers-16-01847-f002:**
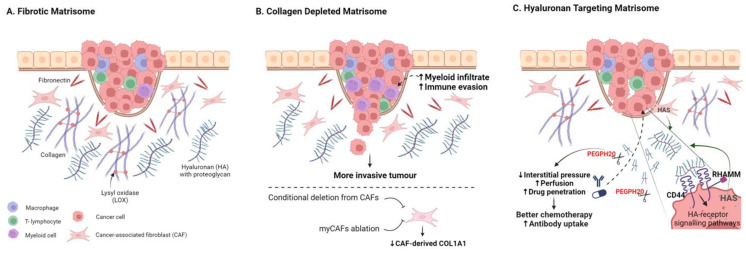
Targeting the Fibrotic Tumour Matrisome. (**A**) Schematic highlighting key abundant matrisome components. (**B**) Targeting stromal-cell-derived collagen 1 fibrosis through conditional deletion of Col1a1 in CAFs or through CAF ablation can result in more invasive tumours and reduced anti-tumour immunity. (**C**) Targeting hyaluronic acid can enhance drug perfusion and therapy response in many pre-clinical models, with numerous trials underway to assess clinical utility. Created with BioRender.com.

**Figure 3 cancers-16-01847-f003:**
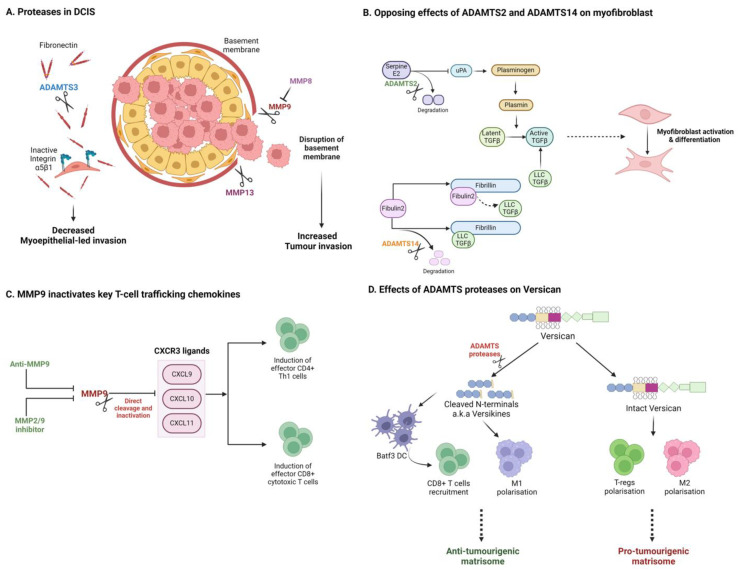
Matrix proteases can impact tumour progression through diverse and opposing mechanisms. (**A**) In DCIS, distinct proteases promote or suppress tumour invasion, dependent on their respective substrates. (**B**) Closely related ADAMTS isoforms can elicit opposing effects on TGFβ and CAF activation in PDAC. (**C**) Proteases can alter the immune microenvironment by degrading key cytokines and chemokines. (**D**) Degradation of matrix components can generate tumour and immune modulating matrikines such as versican-derived versikine. Created with BioRender.com.

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
