# Peer review of "Targets in the Tumour Matrisome to Promote Cancer Therapy Response"

_cancers, 2024, doi:10.3390/cancers16101847_

Round 1
Reviewer 1 Report
Comments and Suggestions for Authors
This review is reasonably written that highlights that targeting the tumour microenvironment (TME) is a double-edged sword, having to strike a balance between breaking down the matrix to allow ingress of immune cells, while at the same time possibly facilitating tumour progression. It is well argued and up-to-date. My comments are mainly grammatical issues, particularly the inappropriate use of commas that have tended to break-up the meaning of many statements, and also the absence of definitions for many abbreviations of some of the less common molecules. In particular:
1. Table 1 in the Name column: some have just the author, others the first author and an indication of the TME component being investigated, while yet others have just the molecule being investigated – be consistent. The tumour type is also unclear, e.g. BRCA, what type of breast cancer; COAD, presumably colonic adenocarcinoma, but again what type (differentiation status). Later on, line 135, the authors use CRC (again no consistency in nomenclature). STAD, presumably stomach adenocarcinoma: intestinal, diffuse?? It would be best to define all abbreviations, particularly the less common ones such as ‘ITGAV’, KPC, KC, ADAMTS, VCAN. HAS in Fig.1- is this hyaluronic acid or hyaluronidase?
2. Line 124: remove comma after ‘provide’
3. Line 188: remove comma after ‘largely’
4. Line 211: remove comma after’ study’
5. Line 284: remove comma after ‘production’
6. Line 371: remove comma after ‘Sparing’
7. Useful to use Greek letters in e.g. aSMA, integrin chains
8. Line 346: ‘unsuccessfully’ should be ‘unsuccessful’
9. Line 364: should be ‘an unmet clinical demand’
10. Line 369: should be ‘promote an integrin alpha 5 beta1-directed…’
11. Line 423: should be ‘…. mice having reduced tumour..’
12. Line 483: ‘indicate direct functionality impact the ability..’ Sense???
13.The review mentions ‘impact’ numerous times, this is a modern ambiguous word. Please indicate what the changes are, are they positive (enhancing) or negative (inhibitory)?
Comments on the Quality of English Language
This review is reasonably written that highlights that targeting the tumour microenvironment (TME) is a double-edged sword, having to strike a balance between breaking down the matrix to allow ingress of immune cells, while at the same time possibly facilitating tumour progression. It is well argued and up-to-date. My comments are mainly grammatical issues, particularly the inappropriate use of commas that have tended to break-up the meaning of many statements, and also the absence of definitions for many abbreviations of some of the less common molecules. In particular:
1. Table 1 in the Name column: some have just the author, others the first author and an indication of the TME component being investigated, while yet others have just the molecule being investigated – be consistent. The tumour type is also unclear, e.g. BRCA, what type of breast cancer; COAD, presumably colonic adenocarcinoma, but again what type (differentiation status). Later on, line 135, the authors use CRC (again no consistency in nomenclature). STAD, presumably stomach adenocarcinoma: intestinal, diffuse?? It would be best to define all abbreviations, particularly the less common ones such as ‘ITGAV’, KPC, KC, ADAMTS, VCAN. HAS in Fig.1- is this hyaluronic acid or hyaluronidase?
2. Line 124: remove comma after ‘provide’
3. Line 188: remove comma after ‘largely’
4. Line 211: remove comma after’ study’
5. Line 284: remove comma after ‘production’
6. Line 371: remove comma after ‘Sparing’
7. Useful to use Greek letters in e.g. aSMA, integrin chains
8. Line 346: ‘unsuccessfully’ should be ‘unsuccessful’
9. Line 364: should be ‘an unmet clinical demand’
10. Line 369: should be ‘promote an integrin alpha 5 beta1-directed…’
11. Line 423: should be ‘…. mice having reduced tumour..’
12. Line 483: ‘indicate direct functionality impact the ability..’ Sense???
13.The review mentions ‘impact’ numerous times, this is a modern ambiguous word. Please indicate what the changes are, are they positive (enhancing) or negative (inhibitory)?
Reviewer 2 Report
Comments and Suggestions for Authors
although it may be clear to many readers, I would actually start defining what exactly is "the Matrisome", and introducing it properly. There may be different "matrisomes", depending on the source and who defined them. This should be done in the first few paragraphs of the introduction, and it should also be described which genes are included in this set of genes. Some of it is actually covered in table 1 but these gene sets are so vastly different in size and gene numbers that its difficult to recognize any overlaps.
Paragraph 1.1 comes already a bit loser to the missing general definition what we may understand as the Matrisome, but again, its not referring to it explicitly. In my understanding, the "matrisome" is a rather well defined set of genes (or a gene set) which more or less reflects the transcriptomic activity of cancer-associated fibroblasts (CAFs) which is of course partly dependent on the tumor type or subtype, but otherwise, is remarkably similar. Its defined in gen set collections such as the Gene Ontology (GO), Cancer Hallmarks, WikiPathways, KEGG or Reactome etc. and should simply be introduced here: what are the core components, and their functions? (Some of this is touched upon in the abstract, but is not more detailed in introduction). How many genes are we talking about? Are there different Matrisomes for different tissues/cancer types?
Next step is the set of gene introduced in Table 1: they are widely different, I suppose, is there a significant overlap between the sets, what are the core genes? Is there any common ground between these stromal signatures? (Venn diagrams, for example, could be used).
I know, this is partly discussed and examples are mentioned starting from line 90 in the manuscript. But figures also help to make a manuscript more pleasurable to read.
Speaking of Figures, Fig. 1 is quite satisfying, although some elements shown (in very small print) are not explained, such as RHAMM or PEGPH20, or HAS. (PEGH20 description follows only in line 271).
I have little criticism to the tumor immune environment chapter (1.2) and the role of stromal genes in immune evasion and immune checkpoint inhibitor responses. Thats nicely covered.
Whats maybe missing are the explicit insights from a growing number of single-cell RNA sequencing studies, investigating the connection between immune cell populations and stromal components, such as different CAF subpopulations (myCAFs, iCAFs, etc).. This emerging topic research has so rapidly gained relevance that this should be more explicitly covered and a few of the better, larger scale studies should be included here.
Chapter 2.1 could actually be placed BEFORE the immune microenvironment chapter as it is or more basic nature and if feels natural that this is part of a more basic introduction before the authors shift to more specific effects of fibrotic/desmoplastic stroma on the immune response. It provies background for some of these effects observed in clinical trials AFTER the immune cell composition is described.
The chapter on the role of hyaluronic acid (HA) is an important one and also harbors a lot of relative novelty; I dont think this has become common knowledge as of yet and the functional roles of HA (and LOX in contrast) are truly interesting and important.
The rest of the review repeats more common positions that have been reviewed many times in other places, such as the role of TGF beta signaling, and of course the matrix (metallo-)proteases as working horses for matrix turnover and remodeling. There are a million articles on the matrix proteases and its difficult to tell something new.
But chapter 3 brings up new ground again, which is related to elements explained in chapter 1 (= tumor immune microenvironment or TIME). One more reason to consider changing the order of topics in the articles, so that the immune checkpoint and the regulation of immune cell components by matrix turnover would be closer connected?
The authors also mention recent and ongoing clinical trials in this chapter, maybe it could be considered to add a table that summarizes some of the more relevant and interesting of these trials; this may be interesting for many readers who like to see the clinical relevance of the topic and the link to drug discovery.
The rest of the article then focuses on the modulation of immune cells, and this section contains maybe the most recent set of references and is therefore also timely and relevant for the field. Of course, this part of the review will not age well, as the field is in constant turmoil and novel finding pop up every day.
Author Response
See attached word document
